# Suppression of Charge Recombination by Auxiliary Atoms in Photoinduced Charge Separation Dynamics with Mn Oxides: A Theoretical Study

**DOI:** 10.3390/molecules27030755

**Published:** 2022-01-24

**Authors:** Yu Ohnishi, Kentaro Yamamoto, Kazuo Takatsuka

**Affiliations:** Fukui Institute for Fundamental Chemistry, Kyoto University, Sakyou-ku, Kyoto 606-8103, Japan; ohoo.suke.you.your.you@gmail.com (Y.O.); kyamamoto.sci@gmail.com (K.Y.)

**Keywords:** charge separation, charge recombination, suppression of charge recombination, substituent effect, nonadiabatic electron dynamics, electron transfer, proton transfer

## Abstract

Charge separation is one of the most crucial processes in photochemical dynamics of energy conversion, widely observed ranging from water splitting in photosystem II (PSII) of plants to photoinduced oxidation reduction processes. Several basic principles, with respect to charge separation, are known, each of which suffers inherent charge recombination channels that suppress the separation efficiency. We found a charge separation mechanism in the photoinduced excited-state proton transfer dynamics from Mn oxides to organic acceptors. This mechanism is referred to as coupled proton and electron wave-packet transfer (CPEWT), which is essentially a synchronous transfer of electron wave-packets and protons through mutually different spatial channels to separated destinations passing through nonadiabatic regions, such as conical intersections, and avoided crossings. CPEWT also applies to collision-induced ground-state water splitting dynamics catalyzed by Mn_4_CaO_5_ cluster. For the present photoinduced charge separation dynamics by Mn oxides, we identified a dynamical mechanism of charge recombination. It takes place by passing across nonadiabatic regions, which are different from those for charge separations and lead to the excited states of the initial state before photoabsorption. This article is an overview of our work on photoinduced charge separation and associated charge recombination with an additional study. After reviewing the basic mechanisms of charge separation and recombination, we herein studied substituent effects on the suppression of such charge recombination by doping auxiliary atoms. Our illustrative systems are X–Mn(OH)_2_ tied to *N*-methylformamidine, with X=OH, Be(OH)_3_, Mg(OH)_3_, Ca(OH)_3_, Sr(OH)_3_ along with Al(OH)_4_ and Zn(OH)_3_. We found that the competence of suppression of charge recombination depends significantly on the substituents. The present study should serve as a useful guiding principle in designing the relevant photocatalysts.

## 1. Introduction

Conversion of photo energy to chemical counterparts is one of the crucial processes in biological systems and sustainable developments in human society. Photosynthesis in plants is made possible by production of protons and electrons extracted from water molecules split with the Mn_4_CaO_5_ complex in PSII [1,2,3,4,5,6,7,8,9,10,11], while artificial photocatalytic systems make use of photoexcitation, along with various chemical and physical technologies [12,13,14,15,16,17]. In artificial charge separation, in particular, a recombination of thus created positive and negative charges quite often follows the separation dynamics, thereby deteriorating the efficiency of charge separation. This paper is concerned with how to suppress charge recombination associated with an elementary process of charge separation due to photoexcited concerted transfers of protons and electrons catalyzed by Mn oxides.

Our studied system is composed of catalytic Mn oxides as a donor of protons and electrons, while as an acceptor of them, organic compounds are adopted. The dynamical mechanism here has been identified and referred to as coupled proton and electron wave-packet transfer (CPEWT) [18,19,20]: upon photoexcitation, Mn oxides create an excited-state electron wave-packet of biradical nature, and one of the radical electrons is transferred to an acceptor, synchronizing a proton transfer from H_2_O attached on the Mn cluster. This kind of proton transfer takes place by passing across quasi-degenerate conical intersections. Therefore, in order to comprehend the relevant elementary dynamics of charge separation, one needs to capture the essential feature of nonadiabatic electron wave-packet dynamics in the excited-state proton transfer, which should be clearly distinguished from ground-state proton transfer and hydrogen atom migration [18,21,22,23].

CPEWT was found to be among the general mechanisms of charge separation. Indeed, we have shown that the charge separation processes driven in the water splitting catalytic cycle by the Mn_4_CaO_5_ cluster in photosystem II (PSII) is materialized by the *ground-state* CPEWT [24]. This charge separation is not due to direct photoabsorption at the Mn_4_CaO_5_ cluster, but is initiated by collision between a cationic molecule and an electron acceptor that is hydrogen-bonded to Mn_4_CaO_5_ [25]. The present process is referred to as chemi-charge separation [25], since it is, in some way, analogous to chemiluminescence (or bioluminescence), in which a chemical reaction starts from ground-state reactants and ends up with a photoluminescent excited state, passing through a conical intersection [26].

In this paper, with respect to the above photoinduced charge separation dynamics, we concentrate on the mechanism of associate charge recombination and the substituent effects on the efficiency of suppressing the recombination, in which auxiliary atoms are doped into the Mn oxides [20]. The minimal model systems as a proton–electron donor in our comparative study are OH-MnOH_2_ and X–MnOH_2_ with substituent groups X containing a doped auxiliary atom in each. *N*-methylformamidine, which is an analog of arginine, is adopted as an acceptor of electrons and protons. In our earlier paper [20], we proposed a basic mechanism of charge recombination dynamics against the charge separation by OH-MnOH_2_ and showed that the relevant charge recombination is significantly suppressed in the similar charge separation by (OH)_3_Ca-MnOH_2_. In this paper, we survey the roles of the doped atoms rather extensively by replacing the Ca atom to the other atoms belonging to the second group of the periodic table, Be, Mg, Sr. We chose these atoms because we were inspired by the comparative studies on whether the clusters of Mn_4_MgO_5_ and Mn_4_SrO_5_ could work as well as Mn_4_CaO_5_ in PSII [27,28,29,30]. Besides these, we surveyed the suppression competence of Zn and Al atoms, because their presence on the earth are relatively abundant. Although the present study is limited in various aspects, we hope the background ideas behind charge separation and recombination, along with the computational methodology, serve as guiding principles for the better design of artificial photocatalysts.

The key mechanisms of the present charge separation and recombination were investigated with quantum dynamics based on nonadiabatic electron wave-packets. The nonadiabatic interactions emerge from the nuclear kinematic interaction with electrons through the quantum nuclear momentum operator [31,32,33,34]. The study of nonadiabatic interaction was initiated as early as in 1934 by Landau, Zener, Stueckelberg, as part of atomic collision physics [32]. It was therefore a one-dimensional theory. Donald Truhlar was among the first who recognized that nonadiabatic interactions are critically important for the studies on chemical reactions, which are generally not of one-dimensional nature. He has long been leading in chemical dynamics by proposing epoch-making theories along with important applications [34,35,36] (it is impossible to make a complete list of his works here). Most of theories of nonadiabatic transitions, and those by the Truhlar group as well, rest on the so-called Born–Huang representation [37], in which time-dependent dynamics of molecules are considered exclusively in the nuclear wave functions, which are supposed to run on the so-called static potential energy hyper-surface (PES) created by time-independent electronic wave functions. Our approach in this paper is, in contrast, based on the *time-dependent* electronic wave functions, and the nuclear kinematic interactions are taken into account through the coupling of electronic wave-packets [38,39,40,41,42]. In this representation, electron wave-packets induced by photoexcitation are *directly* tracked in a real-time scale.

This paper is organized as follows. Section 2 shows the basic results of the present charge separation along with the theoretical method used. Section 3 presents the mechanism of charge recombination and the substituent effects to suppress the recombination. This paper concludes with some remarks in Section 4.

## 2. Dynamics of Charge Separation

### 2.1. Molecular Systems

Our studied molecular systems are X–MnOH_2_ with X being a substituent in which an auxiliary atom is doped. X is chosen to be OH, Be(OH)_3_, Mg(OH)_3_, Ca(OH)_3_, Sr(OH)_3_, Al(OH)_4_, and Zn(OH)_3_. X–MnOH_2_ serves as an electron and proton donor (EPD) to *N*-methylformamidine as an electron and proton acceptor (EPA). Our former studies have shown that the mechanism for charge separation does not depend on the choice of EPA of this type aside quantitative variation [19,20].

Figure 1 displays two schematic molecular compositions as examples for X=OH and X=Sr(OH)_3_. An EPD and the EPA will share a to-be-transferred proton, denoted as H_T_. Both molecules seem to sit on a common plane, but actually they do not.

### 2.2. Global Features of the Potential Energy Surfaces and Excited State
Dynamics

Figure 2 schematically shows the sequence of elementary processes of the present photoinduced charge separation along the coordinate of the H^T^ transfer. In this figure, two elementary processes are illustrated: (i) photoabsorption from state (a) to (b), and (ii) coupled proton and electron wave-packet transfer from state (b) to (d), passing through a conical intersection at (c). The potential energy curves in reality are much more complicated due to the quasi-degeneracy created by the Rydberg-like states of the nitrogen atoms in the acceptor (to be shown later in Section 2.4.2).

To real-time track the electronic wave-packet states involved in the chemical reactions, we used an unpaired electron density *D*(**r**), which is defined as [43]
(1)$$D\left(r\right)=2\rho \left(r,r\right)-\int dr^{\prime }\rho \left(r,r^{\prime }\right)\rho \left(r^{\prime },r\right)$$
along a path. Here, $\rho (r,r^{\prime })$ is the first order spinless density matrix in the coordinate representation. This quantity has proved useful in locating the spatial distribution of unpaired electrons (radicals) [44]. The yellow crowd-like structure represents the distribution of unpaired electron density in Figure 2.

### 2.3. Computational Background

#### 2.3.1. Theory of Nonadiabatic Electron Wave-Packet Dynamics

In our treatment of electron wave-packet dynamics along with nuclear path approximation, the theory generally starts with the path-branching representation [23,40,41,45,46]. However, in light of a particular nature of the problem that we address, in which the relevant nonadiabatic electron dynamics is supposed to propagate in quasi-degenerate electronic state manifolds and the dynamical calculations are pursued only up to 120 fs, the so-called semi-classical Ehrenfest theory (SET) [47,48,49,50,51] is an easier choice for the present applications. The SET can be formally derived as a mean-field approximation to the path-branching representation theory [40,41,45,46]. Another reason for the use of SET is that the aim of the present study is not at highly accurate quantitative reproduction of the results, but rather a semi-quantitative analysis is sufficient. Yet, we have already shown elsewhere that the SET can reproduce an accurate nonadiabatic transition amplitude as long as a single passage is treated [45].

Below, we briefly outline the theory since extensive reviews are found elsewhere [40,41]. The electronic wave-function $\Psi_{\mathrm{elec}}\left(r,t;R\left(t\right)\right)$ at a nuclear position R(t) is expanded in basis functions $\left\{\Phi_{I}\left(r;R\right)\right\}$, such as the Slater determinants and configuration state functions (CSF) in such a manner that
(2)$$\Psi_{\mathrm{elec}}\left(r,t;R\left(t\right)\right)=\sum_{I}C_{I}\left(t\right)\Phi_{I}\left(r;R\right){|}_{R=R(t)},$$
in which $C_{I}\left(t\right)$ is the *I*th time-dependent coefficient to be evaluated. r and R denote the collective representation of all electronic and nuclear coordinates, respectively. The coupled equations of motion for the electron wave-packet along the path R(t) are expressed as
(3)$$i\hbar \frac{dC_{I}}{dt}=\sum_{J}\left[H_{IJ}^{\left(\mathrm{el}\right)}-i\hbar \sum_{k}\frac{dR_{k}}{dt}X_{IJ}^{k}-\frac{\hbar^{2}}{4}\sum_{k}\left(Y_{IJ}^{k}+Y_{JI}^{k*}\right)\right]C_{J},$$
in which the matrix elements are evaluated as
(4)$$H_{IJ}^{\left(\mathrm{el}\right)}=\langle \Phi_{I}\left|{\hat{H}}^{\left(\mathrm{el}\right)}\right|\Phi_{J}\rangle ,\;\;\;\;X_{IJ}^{k}=\langle \Phi_{I}\left|\frac{\partial }{\partial R_{k}}\right|\Phi_{J}\rangle ,\;\;\;\mathrm{and}\;\;Y_{IJ}^{k}=\langle \Phi_{I}\left|\frac{\partial^{2}}{\partial R_{k}^{2}}\right|\Phi_{J}\rangle .$$

The operator ${\hat{H}}^{\left(\mathrm{el}\right)}$ denotes the electronic Hamiltonian at R. The bra–ket notation here is defined as an integral over the electronic coordinates only. Any basis set $\left\{\Phi_{I}\left(r;R\right)\right\}$ can be adopted to give invariant results, as long as they are complete and all the matrix elements in Equation (Equation 3) are correctly taken into account. Although the second-order differential terms $Y_{IJ}^{k}$ in Equation (Equation 3) are the nontrivial corrections to the conventional SET [40,41], they are usually neglected in practical computations because it is multiplied by the small quantity $\hbar^{2}$.

The nuclear paths R(t) are driven by the force matrix $F_{IJ}^{k}$ expressed as
(5)$$F_{IJ}^{k}=-\frac{\partial H_{IJ}^{\left(\mathrm{el}\right)}}{\partial R_{k}}-\sum_{K}\left(X_{IK}^{k}H_{KJ}^{\left(\mathrm{el}\right)}-H_{IK}^{\left(\mathrm{el}\right)}X_{KJ}^{k}\right)+i\hbar \sum_{l}\frac{dR_{l}}{dt}\left[\frac{\partial X_{IJ}^{l}}{\partial R_{k}}-\frac{\partial X_{IJ}^{k}}{\partial R_{l}}\right].$$

The off-diagonal elements of $F_{IJ}^{k}$ are supposed to induce path branching at every single time step. However, such infinite path-branching is actually impossible to manage. Therefore, various applications have been proposed for easier treatment [40,41,45]. The SET is among the simplest and most drastic approximations, in which the wave-packet-average of the force matrix is made as
(6)$$\langle F_{k}\rangle =\sum_{IJ}C_{I}^{*}F_{IJ}^{k}C_{J}=-\sum_{IJ}C_{I}^{*}\frac{\partial H_{IJ}^{\left(\mathrm{el}\right)}}{\partial R_{k}}C_{J}-\sum_{IJ}C_{I}^{*}\left(X_{IK}^{k}H_{KJ}^{\left(\mathrm{el}\right)}-H_{IK}^{\left(\mathrm{el}\right)}X_{KJ}^{k}\right)C_{J}.$$

For a complete basis set $\left\{\Phi_{I}\left(r;R\right)\right\}$ Equation (Equation 6) can be rewritten in the form of Hellmann–Feynman force, such that
(7)$$\langle F_{k}\rangle =-\langle \Psi_{\mathrm{elec}}\left(r,t;R\left(t\right)\right)\left|\frac{\partial {\hat{H}}^{\left(\mathrm{el}\right)}}{\partial R_{k}}\right|\Psi_{\mathrm{elec}}\left(r,t;R\left(t\right)\right)\rangle .$$

#### 2.3.2. Computational Details

To track the real-time nonadiabatic wave-packet dynamics for the present systems, the CISD/RHF level of calculation is employed. Quantum chemical calculations are performed with the GAMESS programming package [52,53]. The atomic basis set chosen is Stevens, Basch, Krauss, Jasien, Cundari effective core potentials (SBKJC ECPs) [54] for Mn, Pople’s 6–31G for (X, O, and C), and 6–31++G for (N, H). The diffused functions included in the 6–31++G basis set are definitely necessary to describe the electron-accepting dynamics, particularly for the Rydberg-like states on the nitrogen atoms [18,19]. We limited our CISD active space to HOMO–(HOMO+80) to obtain the CSFs, the number of which amounts to 3321. With preliminary numerical studies prior to the extensive and costly production, we sought for the size of the active space and the number of configurations small enough to be manageable and large enough to give stable results, with respect to the change of the computational size. Moreover, we performed RHF-level calculations for preliminary geometry optimization, vibrational analysis, and path sampling in the ground state in advance of the full dynamics calculations.

#### 2.3.3. Basic Molecular Orbitals

Prior to the study of electron dynamics, we surveyed the basic electronic structure viewed from the ground state of each system.

We began with the energy level of molecular orbitals for each combined EPD and EPA system. As seen in Figure 3, in which the computed energy levels are tabulated, the highest occupied MO (HOMO) and lowest unoccupied MO (LUMO) are well separated with about 5 eV gaps for all systems. Note however that the HOMO–LUMO gap is indeed far larger than the lowest excitation energy of the CI wave-function of the single–double level. Moreover, we will not photoexcite from HOMO to LUMO due to the very small oscillator strength. Another characteristic feature of the energy levels is the existence of rather dense quasi-degenerate bands in each system. The energy spectra of those bands depend on the choice of the substituent groups, X, of different doped atoms. The quasi-degeneracy emerges from the so-called Rydberg states (diffused quasi-degenerate states) centered on the nitrogen atoms of *N*-methylformamidine, which is expected to serve as a large electron accepting site. The similar quasi-degeneracy also shows up in the excited states of relevant electron transfer as will be demonstrated later in Section 2.4.2, which makes nonadiabatic electronic wave-packet calculations very complicated.

We next surveyed the spatial distributions of the relevant molecular orbitals in the order of HOMO–(LUMO+2) for a systematic comparison among those for Be(OH)${}_{3}$ (Figure 4), Mg(OH)${}_{3}$ (Figure 5), and Sr(OH)${}_{3}$ (Figure 6). (The similar molecular orbitals for X=OH and X=Ca(OH)${}_{3}$ are presented in [20].) Those of X=Be(OH)${}_{3}$, Al(OH)${}_{4}$, and Zn(OH)${}_{3}$ are not shown graphically, yet they have characteristics more-or-less similar to those of Figure 4, Figure 5 and Figure 6. These diagrams share significant features. We first note that MOs from HOMO (and HOMO-1 and HOMO-2 as well, not shown here graphically) are localized in the EPD moiety (Mn-oxide complex), while LUMO and LUMO+1 tend to localize in the EPA site (*N*-methylformamidine). In particular, all the LUMOs shown here are made up with the Rydberg-like diffused orbitals on the N atoms. Such diffused vacant orbitals of LUMO suggest that they should serve an efficient electron acceptor. These common features are consistent with the large and clear energy gaps between HOMO and LUMO seen in Figure 3.

More importantly, the small spatial overlap between HOMO and LUMO (and LOUMO+1 as well) suggests a very small oscillator strength between them. Therefore direct photoexcitation from HOMO to LUMO is expected to rarely take place. In other words, an efficient photoexcitation demands to carefully pick the excited states of large oscillator strengths among those having lowest possible excitation energy.

The final aspect to note is that the orbitals of LUMO+2 of all the systems have a feature similar to each other, which extend over both EPD and EPA, thereby suggesting electron delocalization entirely over all the EPD and EPA in rather coherent and oscillatory manner.

### 2.4. Coupled Proton and Electron Wave-Packet Dynamics (CPEWT) in Excited
States

#### 2.4.1. Running Nonadiabatic Electron Wave-Packet Dynamics

The choice of the initial conditions for both electron wave-packets and nuclear path dynamics were made as follows [19]. Regarding the ground state of the present systems, it was found that RHF is good enough to approximate the CI energy. Thus, we calculated normal modes with the RHF PES and determine the zero point vibrational energy for each mode. In each one-dimensional phase space, we randomly chose a phase space point (position and velocity) at each time of sampling. An initial condition for the nuclear path is prepared in this way. Electron wave-packets were sampled among such random nuclear positions and lifted up to photoexcited states. An electronic wave-packet created in the excited state manifold is prepared in such a way that it has equal weights (with no bias) among those adiabatic excited states that lie in the range of excitation energy of 3.0–3.5 eV and bear the oscillator strengths $f_{0n}$ larger than 0.1. Our preliminary study shows that the qualitative aspect presented below about charge separation and the state-mixing does not depend much on the choice the initial sampling weights among those selected states, since the natures of those selected excited states involved herein are somewhat similar to each other.

To start the SET dynamics, we assigned the zero point vibrational energy (0.5 ℏω for each normal mode) to each of the nuclear paths. We ran, ab initio, molecular dynamics with the RHF level of calculation to approximately obtain the path of zero-point vibration motion. This approximation is valid since the PES near the potential basin of the ground state is mainly dominated by the RHF ground state configuration. The nuclear sampling points, both positions and momenta, for excited-state dynamics are randomly picked up from these sampling paths. It turns out that the dependence of the charge separation dynamics on the initial condition is not significant, because the present excited-state dynamics is more or less down-sloping in the initial stage. After the excitation thus prepared, both H^T^ and electron wave-packets are spontaneously transferred from EPD to EPA, with different speeds along their own pathways.

#### 2.4.2. Real-Time Tracking of the Dynamics

In all of the systems studied, charge separation triggered by the transfer of H^T^ and associated recombination occur within 120 fs. The proton transition is customarily judged in terms of a graph of the bond-length crossing between H^T^-O (EPD) and H^T^-N (EPA). To visualize (not for wave-packet calculations) the electron wave-packet dynamics, we expand the total electronic wave-packet of Equation (Equation 2) in the adiabatic electronic wave-functions $\left\{\Phi_{n}^{\left(ad\right)}\left(r;R\right)\right\}$ at each geometry along the nuclear path R(t)
(8)$$\Psi_{\mathrm{elec}}\left(r,t;R\left(t\right)\right)=\sum_{n}C_{n}^{\left(ad\right)}\left(t\right)\Phi_{n}^{\left(ad\right)}\left(r;R\right){|}_{R=R(t)},$$
each $\Phi_{n}^{\left(ad\right)}\left(r;R\left(t\right)\right)$ having the electronic energy $V_{n}\left(R\left(t\right)\right)$ (including the nuclear repulsion energy). We first draw a graph $V_{n}\left(R\left(t\right)\right)-V_{0}\left(R\left(t\right)\right)$ as a function of time *t* along a path R(t), which is the relative height of the excited states viewed from the ground state $\Phi_{0}^{\left(ad\right)}\left(r;R\left(t\right)\right)$. Those graphs are shown in Figure 7. Note that the bottom axis in all the panels indicate the reference ground state at each R(t). To demonstrate the population of each adiabatic state; that is ${\left|C_{n}^{\left(ad\right)}\left(t\right)\right|}^{2}$, a blue cloud-like shadow is superposed on the curves $V_{n}\left(R\left(t\right)\right)-V_{0}\left(R\left(t\right)\right)$. The darker it looks, the larger lies the population.

It is immediately noticed from Figure 7, the global feature of the charge separation through the nonadiabatic transitions are rather common. For instance, the case of X=OH, panel (a) of Figure 7 exhibits its first nonadiabatic transition at about 61 fs, and the electronic wave-packet also enters the right valley of Figure 2. We note that the diagrams of Figure 7 are just a one-dimensional projection of the full-dimensional dynamical calculation results, and complicated dynamics should take place in the coordinates transversal to the SET path.

## 3. Charge Recombination


### 3.1. Mechanism of Charge Recombination

We now enter the heart of this paper, which is about the mechanism of charge recombination and the suppression of it as a substituent effect. In Figure 7a are noticed two almost horizontal curves from the bottom, which indicate the potential energy curves of the S1 and S2 states viewed from the ground state. Both are close at the level of about 1.5 eV and thereby quasi-degenerate. Our previous study has unveiled that the S1 and S2 both originate from d–d excited states localized mainly on the Mn atom in the EPD, along which H${}^{T}$ can be guided back to EPD together with the electron wave-packets [19]. Well separated from the states higher than and equal to S3, S1 and S2 begin to run from t=0 to around t=65 fs (Figure 7a), at which these two states cross one of the other excited states, which have already carried much population by the crossing time. Obviously this crossing takes place in the right potential valley of Figure 2 after charge separation. After the crossing, S1 and S2 are seen to earn a dramatic amount of electronic state population.

A simplified path-branching method [19] highlights the essential feature of the dynamics in Figure 12 cited in Reference [20] after branching to two paths at a little earlier time than t=65 fs. It turns out that one path did not undergo nonadiabatic transition to S1 and S2 state and remain in the charge separation manifold, which are about 73% in population, remain in the charge separation state up to 130 fs. On the other hand, the remaining packets of population about 27% have made transition to the second path composed of S1 and S2 states and have returned to the Mn atom along with the back transfer of H${}^{T}$. This is a typical charge recombination in this system. Such a large portion of the charge separation is reduced even by a single passage of this type of crossing. It is therefore not hard to expect that the electronic state population remaining in the charge separation manifold will continue to be lost by another, and consecutive nonadiabatic crossings with S1 and S2.

### 3.2. Suppression of the Charge Recombination

To identify the substituent effect on charge recombination, we compare the panels Figure 7b–g with Figure 7a. First we recall that the energy at t=0 in each graph indicates the initial energy spectra of the states at a geometry. In the panel Figure 7a for the X=OH system, we have two near-degenerate d–d origin states S1 and S2 at about 1.5 eV, while those two states are widely split to the states, now found at around 0.9 eV and 2.0 eV, respectively, in all the other systems of Figure 7b–g. We do not observe a significant difference among them, or, we do not see much dependence on the choice of the doped atoms Be, Mg, Ca, Sr, Al, and Zn in the energy level of S1 and S2 at t=0. The large split between S1 and S2 energies must have been induced by the chelate-like addition of X to Mn than simple -OH bonding (see Figure 1).

As the charge separation proceeds, the new S2 in each system other than X=OH merged into the state manifold of charge separation, while the new S1 remains widely separated with steady energy, around 1.0 eV. The graphs in Figure 7 clearly depict it. Due to the merge of the S2 state to the charge separation group in Figure 7b–g, the pathway of electron flux from the EPD to EPA is significantly modulated in such a way to extend over the moiety of auxiliary atoms (not shown graphically here, but we refer to Figure 9 cited in Reference [20]). Furthermore, the S2 energies are seen to become significantly lower after the first nonadiabatic transitions causing the charge separation, and there should be only a small chance for the state population distributed among those low energy manifold to bring the S2 back to the original situation at t=0. Therefore, only S1 remains as the charge recombination channel back to the d–d state, in a marked contrast to the case of X=OH. Thus the number of the charge recombination channels is reduced to one and the rate should be suppressed significantly. The extent of the charge recombination for the substituted systems is henceforth to be measured in terms of the population of the S1 state for the systems other than X=OH.

Let us perform the simplified path-branching [20] for the case of X=Mg(OH)${}_{3}$ as an illustrative example. In Figure 8, we track two SET paths after the first branching time 75 fs. The path remaining in the valley of charge separation state (see Figure 2) is presented in panel (a) and (b) of Figure 8 (denoted as Group1), while the charge recombination path is illustrated in panel (c) and (d) (Group2). The SET path of Group2 resumes to run with the same adiabatic electronic state coefficients $C_{n}^{\left(ad\right)}\left(t\right)$ of Equation (Equation 8) for the S1 state only and zero otherwise, while for Group1 $C_{n}^{\left(ad\right)}\left(t\right)$ are retained as were except setting the coefficient for the S1 to zero. The crossing between the bond lengths R${}_{OH^{T}}$ and R${}_{NH^{T}}$ in panel (c) clearly indicates that H^T^ of Group2 has returned to the site of EPD (Mn site), while the path in Group1 remains in the proton transfer region. Panel (b) shows that the path of Group1 undergoes nonadiabatic interaction with the S1 state many times. Likewise the S1 state seems to lose the electronic population back to the charge separation state before about 100 fs (see Figure 8d). Note that these individual paths in Figure 8 do not represent the unbranched SET path of Figure 7c or Figure 9c.

We note that all of the present quantum wave-packet calculations were performed under a situation where EPD and EPA kept contact coherently. As long as the coherence is maintained, time-reversal dynamics is possible in principle. Therefore, the role of decoherence (accidentally) introduced to the systems to break the time-reversal should be taken into account. In this sense our discussion covers only an ideal state in the coherent dynamics. For the role of decoherence in the electron wave-packet dynamics and charge separation, see references [24,55].

### 3.3. Competence of Suppressing the Charge Recombination

To quantify how much the electronic population can flow out to the primary channel of charge recombination we calculate ${\left|C_{n}^{\left(ad\right)}\left(t\right)\right|}^{2}$ of S1 and S2 for X=OH, and S1 alone for the others. Figure 9 shows those graphs. It is easy to visibly judge which substituents are effective to suppress the charge recombination. In particular the substituents containing Ca and Sr and Zn are efficient. It is also noticed that those populations undergo time fluctuation. This is because the EPD and EPA system are kept coherently contact and the partial back-transfer of electrons from the S1 to the charge transfer manifold is also allowed.

To make a little more quantitative discussion, we numerate the extent of the charge recombination by integrating the population in the recombination channel $P_{M}\left(t\right)$ for a molecular system M as
(9)$$S_{M}\left(t_{\max}\right)=\int_{0}^{t_{\max}}dsP_{M}\left(s\right),$$
where the sampling time length $t_{\max}$ has been set to 120 fs. Table 1 shows the results. It is clear that, in the second group of the periodic table, the charge recombination is suppressed in the order
(10)$$\left(\mathrm{OH}\right)\ll \;{\mathrm{Be}(\mathrm{OH})}_{3}<{\mathrm{Mg}(\mathrm{OH})}_{3}<\;{\mathrm{Ca}(\mathrm{OH})}_{3}∼{\mathrm{Sr}(\mathrm{OH})}_{3}$$
and $\left\{{\mathrm{Be}(\mathrm{OH})}_{3},\;{\mathrm{Mg}(\mathrm{OH})}_{3},\;{\mathrm{Al}(\mathrm{OH})}_{4}\right\}$ make a weak suppression group, while $\left\{{\mathrm{Ca}(\mathrm{OH})}_{3},\right.$$\left.{\mathrm{Sr}(\mathrm{OH})}_{3},\;{\mathrm{Zn}(\mathrm{OH})}_{3}\right\}$ form a strong group.

The accumulated (charge recombination) population $S_{M}$ defined above does not necessarily reflect the transition probability from S2 to S1. An important note here is that such a nonadiabatic passage can take place many times with different populations on the S2 state, since S2 also keeps nonadiabatically interacting with the higher states. Therefore, the accumulated population leaking to the recombination channel(s) and the associated efficiency of its suppression should be carefully compared depending on the systems and aims under study.

We admit we failed to single out the clear-cut origin of the above difference in the competence of charge recombination among the atoms from the simple features of molecular orbitals of the systems, which have been partially presented in Figure 1. This seems rather natural in view of the very complicated features of the potential surfaces, which lead to frequent electronic state mixing from the states-to-states. Note that the state mixing has already begun as early as right after photoabsorption, even far before the electron and proton transfer (see Figure 7).

## 4. Concluding Remarks

We studied the photoinduced charge separation dynamics and the associated charge recombination in the system of X–MnOH${}_{2}$ tying to *N*-methylformamidine, with X chosen to be OH, Be(OH)${}_{3}$, Mg(OH)${}_{3}$, Ca(OH)${}_{3}$, Sr(OH)${}_{3}$, Al(OH)${}_{4}$, and Zn(OH)${}_{3}$. The mechanism of charge separation was identified to be the coupled proton and electron wave-packet transfer (CPEWT) from the Mn oxide to the acceptor (A) along the proton transfer coordinate across a region of conical intersections. The electronic state thus produced after the transfer of protons and electrons, which is schematically denoted as X–MnOH⋯A^−^H^+^, can undergo another nonadiabatic transition, with some states originating from d–d excitation in X–MnOH_2_ (EPD) and extending to the acceptor moiety. By passing across the second conical intersection(s), the state partially turns to one or some of those excited states of X–MnOH_2_ and can come back to the excited state of X–MnOH_2_. It eventually returns to the ground state by spontaneous photoemission (or nonradiative transition if any) thereby completing the perfect charge recombination.

We have approximately quantified the suppression efficiency of charge recombination in terms of the path-branching method along with the electronic state population on the d–d origin states. The efficiency of charge recombination is found to be in the order of Be(OH)_3_ < Mg(OH)_3_ < Ca(OH)_3_ ∼ Sr(OH)_3_. The competence of Zn(OH)_3_ is similar to those of Ca(OH)_3_ and Sr(OH)_3_, while Al(OH)_4_ falls in the weak group together with Be and Mg.

In the present work, we did not study the quantitative dependence of charge recombination on the EPA. A good choice of the EPA must be crucial in a practical design of efficient charge separation systems. In our previous studies, we observed only a weak dependence of the choices among amino acid residues as an EPA. [19]. The presence of nitrogen atoms in them, with a set of wide-spread Rydberg-like vacant orbitals in appropriate energy levels, is a key for these system to accept electrons well into the capacity. It is conjectured that nature has made use of this property of a nitrogen atom in molecular evolution. However, the present paper has nothing to claim about the relative efficiency for charge recombination among different EPAs. Nevertheless, we should comment here on other important roles for the EPA to play, to control the efficiency of charge recombination. First of all, we note that the charge separation states send temporarily electrons back over the site of the electron donor in coherent electronic motion. This is also suggested by the spatial distribution of molecular orbitals of LUMO + 2 for all the molecules exhibited in Figure 4, Figure 5 and Figure 6, which extend spatially wide covering both EPD and EPA. Therefore as long as the quantum mechanical coherence is maintained between EPD and EPA, such back and forth motion of excited-state electron wave-packets is unavoidable. In this aspect, two clues are found in biological systems, which work to prevent significant charge recombination. The first one is a Y-shaped molecular structure to be implemented to an EPA, in which the channel of an electron transport and that for proton relay transfer are physically branched (in the Y-shape) and, thereby, electrons and protons are to be transported to different (and well separated) destinations [56]. In this regard, we need to take account of the effect of a hydrogen-bonding network of water molecules in between EPA and EPD [57,58]. The second one is an efficient (yet natural) control of coherence and decoherence among the composite subsystems collaborating together in charge transfer. It has been shown theoretically that such appropriate switching-on and -off of coherence can make it possible for a one-way transfer (unidirectional without back flow) of protons and electrons to be realized in practice [55]. The very long time scale of electron transport in PSII suggests [59] that such on-and-off switching should be materialized by the large amplitude motion of (large) protein molecules and/or collision (contact) frequency that triggers the initiation of coherent process of CPEWT [55]. Such an example is seen in the water splitting catalytic cycle by Mn_4_CaO_5_ in photosystem II [24]. This catalytic cycle consists of four time charge separation dynamics, each of which is known to be well separated by relatively long time-intervals [59]. One-way transport of protons and/or electrons is indeed one of the key processes in biological systems.

Before concluding this paper, we would like to detour our discussion about the possible reason for the presence of Ca atom in the Mn_4_CaO_5_ cluster in nature. Experimental studies have found that among possible substitutions with respect to Ca atom, Mn_4_SrO_5_ can work chemically as well as Mn_4_CaO_5_, but Mg cannot [27,28,29,30]. Yet, it is well known that only Mn_4_CaO_5_ actually exists in nature as water-oxidizing complex (WOC) in PSII. The role of Ca atom there has been found to be very different from that of (OH)_3Ca-MnOH_2__ in the above photoinduced charge separation and recombination [24]. Ca in WOC locates in the (skewed) cubic sub-structure of Mn_3_CaO_4_ and is bound by two water molecules. It is suggested that one of these water molecules is utilized in order to keep the position and orientation of WOC in an appropriate direction with respect to the acceptors through hydrogen bonding, and the other one is supposed to be involved in water splitting dynamics directly. Our former study found no clear evidence about the role of suppression of charge recombination in contrast to Ca in (OH)_3_Ca-MnOH_2_ [24]. Nevertheless we may guess that in the old history of molecular evolution on the earth, possible photo-energy conversion associated with water splitting should have started with photoinduced charge separation with simpler Mn oxide complexes. It is quite likely that the Ca atom was naturally chosen as a partner of those Mn complexes to suppress charge recombination. From the functional view of suppressing charge recombination alone, Sr and Zn can be regarded as good candidates, as we found above. However, Ca is far more abundant than them and, thereby, available on the globe surface. It is rather natural to conceive that Ca of such role remains in bi-nuclear Mn oxo complexes and possibly trinuclear Mn oxo complexes in the evolution process before finally reaching the ground-sate charge separation by Mn_4_CaO_5_.

We note that the present calculations have not been about the total efficiency of charge separation for a given (fixed) photon spectrum, such as sunlight. Besides, we could not perform statistical calculations, in which a large number of initial conditions must be prepared. This is simply because the calculations of nonadiabatic electron dynamics are costly. Yet, we note that photoexcited charge separation and recombination in our studied systems are so complicated in the sequence of multidimensional and quasi-degenerate nonadiabatic transitions that simple molecular orbital analysis and one-dimensional classic theories of nonadiabatic transitions, such as the Landau–Zener one, are unlikely to treat well. The present study is not intended to offer the best choice of auxiliary atoms and the combination thereof as the suppressor, rather, the main aim was to show how the photoinduced charge separation, due to the CPEWT, is necessarily followed by charge recombination, by passing across different (and those not known yet) conical intersections, and that such recombination can be controlled by a chemical modification. Since these mechanisms are different from those on the surfaces of solid states, such as silicon crystal and amorphous, the present study should offer different guiding principles in the search of novel and/or efficient photocatalysts for charge separation.

## Figures and Tables

**Figure 1 molecules-27-00755-f001:**
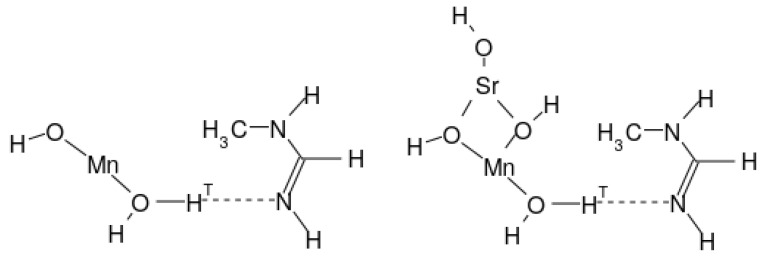
Molecular structures studied for photoinduced charge separation and charge recombination. The basic structure with X=OH (**left**) and a substituted compound at Mn atom (**right**).

**Figure 2 molecules-27-00755-f002:**
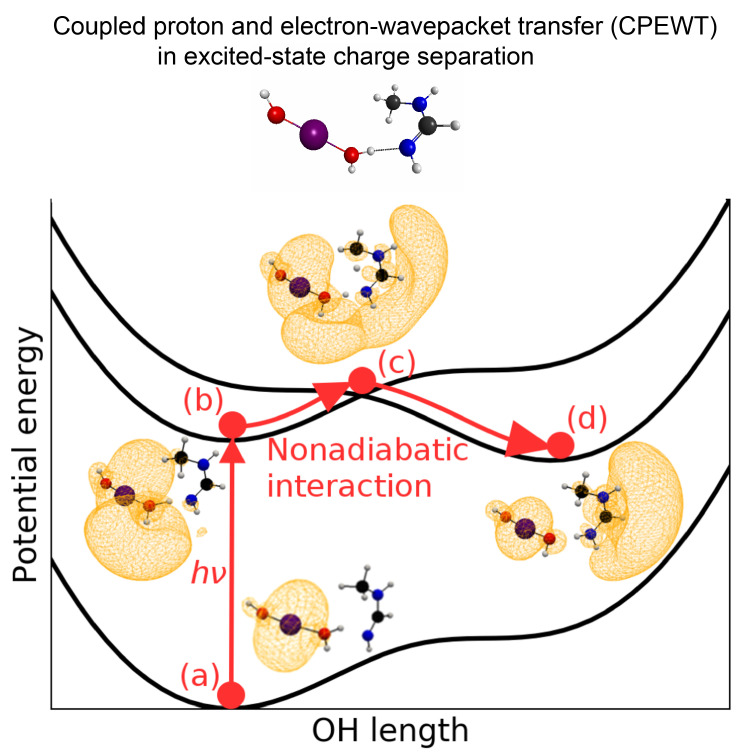
Schematic diagram to illustrate the elementary processes of the present photoinduced charge separation (the excited-state proton transfer) in the combined X–MnOH${}_{2}$ and *N*-methylformamidine system here with X=OH. (**a**) A point on the ground state. (**b**) Photoexcited biradical state. (**c**) Nonadiabatic region on the way of proton transfer. (**d**) State of charge separation completed by the transfer of one of biradical electron wave-packets and H${}^{T}$. The excited states, including the nonadiabatic region, are heavily quasi-degenerate, as a matter of fact. The uphill ground-state potential curve suggests that the ground-state proton transfer does not occur in this system. See the text for the snapshots of electronic states.

**Figure 3 molecules-27-00755-f003:**
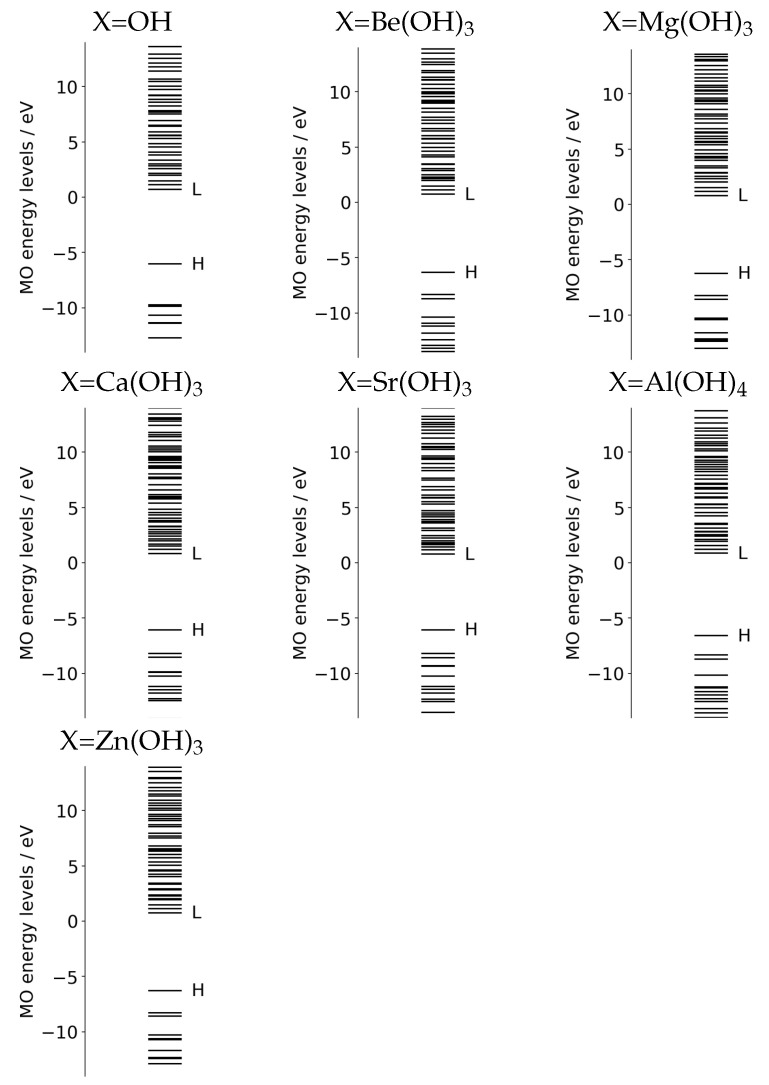
Energy levels of molecular orbitals of each ground state for X–MnOH_2_ combined with *N*-metylformamidine. “H” and “L” respectively, indicate HOMO and LUMO, which are well separated, commonly with about 5 eV gap energy.

**Figure 4 molecules-27-00755-f004:**
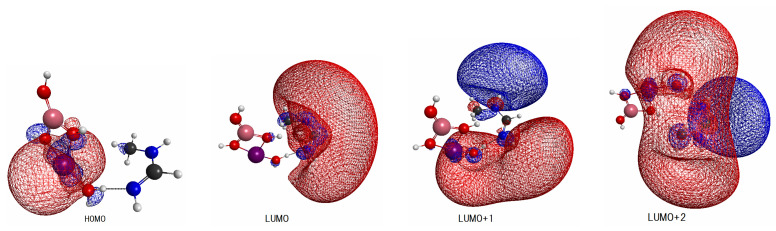
Spatial distribution of molecular orbitals of the levels HOMO–(LUMO+2) for the combined X–MnOH_2_ and *N*-methylformamidine system with X=BeOH_2_. The color assignment of the balls in figure is; purple (Mn), orange (Be), red (oxygen), black (carbon), blue (nitrogen), and gray (hydrogen).

**Figure 5 molecules-27-00755-f005:**
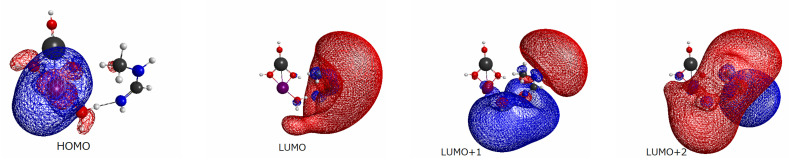
The same as Figure 4 except for X=Mg(OH)_3_. The color assignment of the balls in figure is; purple (Mn), black (Mg), red (oxygen), black (carbon), blue (nitrogen), and gray (hydrogen).

**Figure 6 molecules-27-00755-f006:**
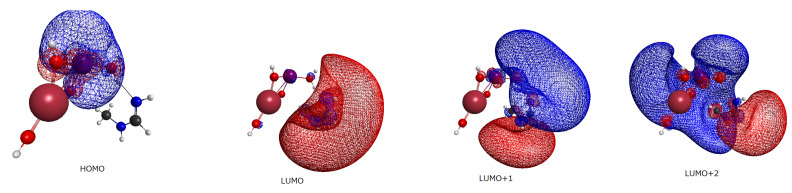
The same as Figure 4 except for X=Sr(OH)_3_. The color assignment of the balls in figure is; purple (Mn), dark red (Sr), red (oxygen), black (carbon), blue (nitrogen), and gray (hydrogen).

**Figure 7 molecules-27-00755-f007:**
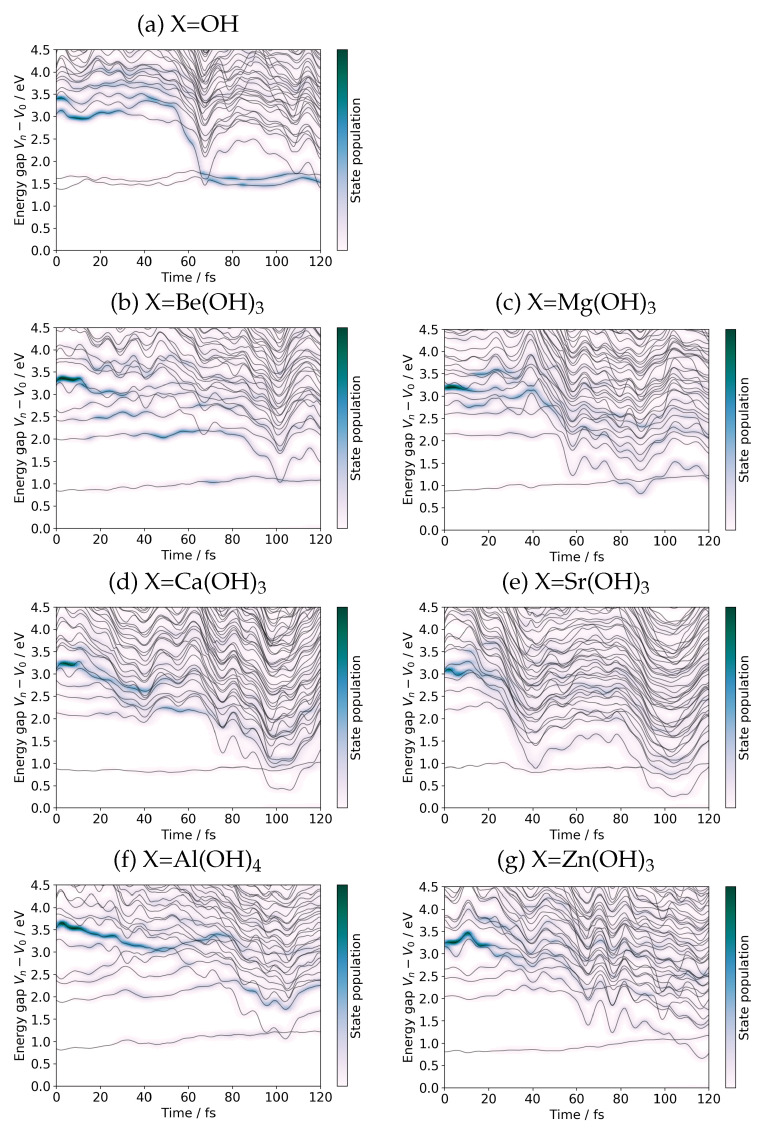
Time variation of $V_{n}-V_{0}$ and ${|C_{n}^{\left(ad\right)}\left(t\right)|}^{2}$ monitored as each CPEWT process proceeds along a SET path for the combined X–MnOH_2_ and *N*-methylformamidine system. The horizontal axis of each graph represents $V_{0}\left(R\left(t\right)\right).$ The timing of proton transfer is: (**a**) X=OH; 61.0 fs, (**b**) Be(OH)_3_; 56.8 fs, (**c**) Mg(OH)_3_: 49.8 fs, (**d**) X=Ca(OH)_3_; 69.2 fs, (**e**) Sr(OH)_3_; 33.4 fs, (**f**) Al(OH)_4_; 56.4 fs, and (**g**) Zn(OH)_3_; 58.0 fs. See Section 2.4.2 for the definition of the timing of proton transfer.

**Figure 8 molecules-27-00755-f008:**
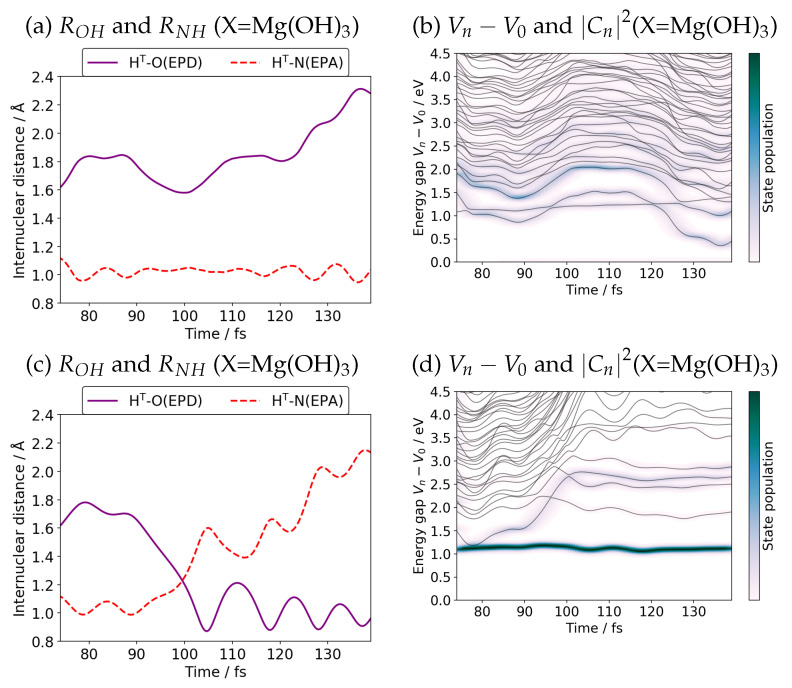
Position of H_T_ and the adiabatic state energies after path branching at branching time about 75 fs for the system of X=Mg(OH)_3_. The upper column lays out Group1 (remaining in the charge separation state), while the lower one for Group2 (charge recombination state). Panels (**a**,**c**) indicate the position of H_T_. The adiabatic potential energy measured from the ground state along the individual branched paths are shown in (**b**,**d**), which are to be compare with Figure 7c.

**Figure 9 molecules-27-00755-f009:**
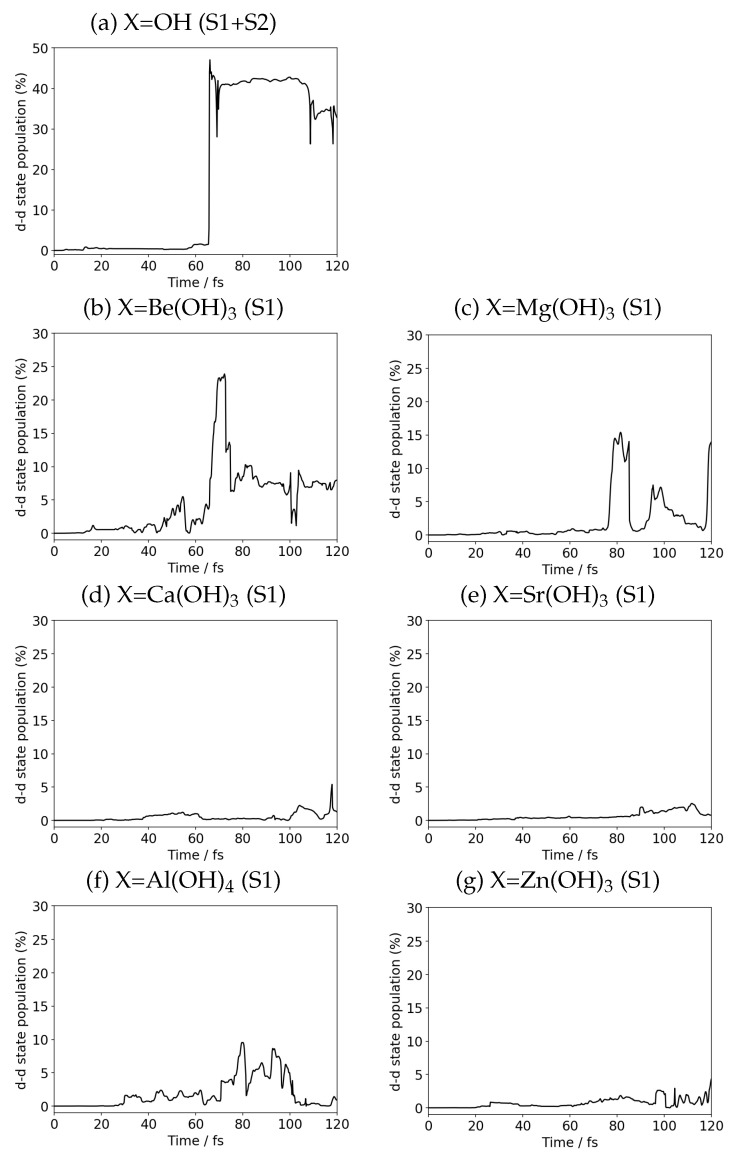
Time variation of the population of charge recombination channel in the combined X–MnOH${}_{2}$ and *N*-methylformamidine system; S1 + S2 states for (**a**) OH, and S1 state only for (**b**) Be(OH)${}_{3}$, (**c**) Mg(OH)${}_{3}$, (**d**) Ca(OH)${}_{3}$, (**e**) Sr(OH)${}_{3}$, (**f**) Al(OH)${}_{4}$, and (**g**) Zn(OH)${}_{3}$.

**Table 1 molecules-27-00755-t001:** Population in the charge recombination states accumulated during the first 120 fs.

X	Population (%·Time)
OH	22.07
Be(OH)${}_{3}$	5.62
Mg(OH)${}_{3}$	2.44
Ca(OH)${}_{3}$	0.59
Sr(OH)${}_{3}$	0.73
Al(OH)${}_{4}$	2.26
Zn(OH)${}_{3}$	0.83

## Data Availability

All data included in this manuscript are available upon reasonable request by contacting the corresponding author.

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
