# Peer review of "Suppression of Charge Recombination by Auxiliary Atoms in Photoinduced Charge Separation Dynamics with Mn Oxides: A Theoretical Study"

_molecules, 2022, doi:10.3390/molecules27030755_

Round 1
Reviewer 1 Report
Please, have a look at the attached pdf.

Reviewer 2 Report
The authors study the substitution effect on the coupled electron-proton transfer in Mn-N-methylformamidine models using Ehrenfest dynamics.
This is a timely study which will be of interest to the community. However, the presentation should be improved taking into account the comments below.
- abstract: "mutually different spatial channels to separate destinations" Actually, it is not clear how this is realized in the present case as electron and protocol take the same path to the acceptor.
- There are several instances where - in my personal view - the style is not appropriate for a scientific paper, e.g.
- page 2 conversion of photo energy is a crucial process in our society
- page 2: CPEWT must be the CT mechanism Mother Nature makes use of (besides this is a very strong statement)
- page 3: The appraisal of Truhlar's work could be tuned down a bit
- page 17: guessed that nature has made good use ...
- page 18: nevertheless we can guess that in far ancient age....
- page 2: "make use of photo excitation in various chemical .. methodologies" The use of methodologies seems a bit odd here
- page 2: second paragraph, there is something missing in the first sentence
- page 4, after eq. 4: "practical methods have been proposed for practical applications"
- page 4: preliminary studies are quoted to confirm the used level of theory. More details are needed to make clear for which quantities this is a good compromise
- page 5: last sentence before Sec. 2.4 sounds a bit odd. Do you mean that the molecules are not planar?
- page 5: I don't get why the single particle MOs are discussed a length, even though it is said that the real state energies are much different. In this respect it is not clear why should one look at the MOs given in Fig. 3 - 7.
- page 6: first sentence "excited states of relevant electron transfer" a bit vague. Besides, the following statement that the calculations are difficult - I thought the fact that the curves run parallel is the condition for performing meaningful Ehrenfest dynamics anyway.
- page 8: I suggest to have such a mechanistic picture earlier in the paper, the introduction of the model would clearly benefit. But, I didn't get why this reaction runs uphill. Also I wonder that's the relation to the S1/S2 states discussed below. In caption a)-d) should be mentioned.
- page 11: How about ZPE leakage? How many sampling points? In this respect is there any convergence?
- page 11: after Eq 8, "bottom straight lines" Do you mean the axis?
- page 11, last sentence. "complications unseen take place" Remarkable sentence!
- discussion fo Fig. 9: From the figures it is not clear how they related to proton transfer. there is not distance info as claimed in the text. Further the statement on page 12 that "crossing takes place in right valley" is a bit hard to follow as no info is provided.
- page 13: Title: What's "another" and also "d-d" was not used before and comes a bit out of the blue
- For the whole S1/S2 discussion it would be helpful to have the characters of the states along the trajectory. As the cross with other states it is obvious that they won't be of pure d-d character, which will play a role for the mechanism.
- page 13, last sentence: "energy lowering suggests the recombination via S2 is unlikely": don't get it, S1/S2 come very close and even cross later on. Also S2 energy goes up again....
- page 14: second paragraph: The path remaining...:" sentence
- page 14: eq. 9: I wonder what is special about the final time 120 fs???
- page 15: Table 1: 3 digits are most likely not relevant
- page 18: top "It is likely...." switching on/off of coherence is a rather bold statement. "Large amplitude motion of large proteins" Not clear, I would have guessed that local fluctuation dynamics is essential for decoherence.
- page 18 "surface hopping model ..highly unlikely to be applicable" This is pure speculation and as such a rather bold statement.
- page 18: Next sentence "We believe ..." I don't like the term believe in a scientific work. Besides, given the rather approximate nature of the present model, this is again a rather bold statement.
- page 18: "another class of CIs" there was no mentioning of classes of CIs so far?
